# The Strong Inhibition of Pancreatic Lipase by Selected Indonesian Medicinal Plants as Anti-Obesity Agents

**DOI:** 10.3390/cimb47010039

**Published:** 2025-01-09

**Authors:** Min Rahminiwati, Dyah Iswantini, Rut Novalia Rahmawati Sianipar, Rani Melati Sukma, Susi Indariani, Anggia Murni

**Affiliations:** 1School of Veterinary Medicine and Biomedical Sciences, IPB University, Bogor 16680, West Java, Indonesia; minrahminiwati@gmail.com; 2Tropical Biopharmaca Research Center, IPB University, Bogor 16128, West Java, Indonesia; trivadila@apps.ipb.ac.id (T.); susiin@apps.ipb.ac.id (S.I.); anggia_murni@apps.ipb.ac.id (A.M.); 3Department of Chemistry, Faculty of Mathematics and Natural Sciences, IPB University, Bogor 16680, West Java, Indonesia; rutnovaliasianipar@apps.ipb.ac.id (R.N.R.S.); rani_melatisukma@apps.ipb.ac.id (R.M.S.)

**Keywords:** Indonesia, natural resources, obesity, pancreatic lipase enzyme

## Abstract

Obesity is characterized by the accumulation of excessive fat, potentially leading to degenerative diseases. Pancreatic lipase, an enzyme responsible for converting 50–70% of dietary fat into monoglycerides, free fatty acids, and various other smaller molecules, plays a crucial role in fat metabolism. Therefore, this study aimed to review selected Indonesian medicinal plants with the potential to inhibit the activity of the pancreatic lipase enzyme. The results showed that kunci pepet (*Kaempferiae angustifolia* Rosc.), asam gelugur (*Garcinia atroviridis*), temulawak (*Curcuma xanthorrhiza*), jombang (*Taraxacum officinale* F. H. Wigg), pegagan (*Centella asiatica*), and pala (*Myristica fragrans*) had strong inhibitory effects, exceeding 50% for both in vitro and in vivo studies. Therefore, further studies are needed to explore the potential of these medicinal plants as anti-obesity treatments.

## 1. Introduction

Obesity is attributed to a rise in fast food intake, unhealthy lifestyles, psychological factors, socioeconomic level, diet programs, age, gender, and insufficient physical exercise. The imbalance between incoming and outgoing energy causes excessive fat accumulation, leading to a body weight significantly above normal [1,2].

Apart from being a health problem, obesity is also a social problem, because some people with obesity are vulnerable to discrimination. The number of people with obesity is increasing rapidly almost all over the world. This has led to focused attention from various health institutions to reduce its prevalence. In 2022, the WHO reported that 43% (2.5 billion) of adults worldwide were classified as overweight, with approximately 16% (890 million) suffering from obesity. Furthermore, 37 million children under 5 years old, 390 million children and adolescents aged 5–19 years old, as well as 160 million adults are obese. The WHO defines obesity based on body mass index (BMI), computed by dividing body weight in kilograms (kg) by height^2^ (m^2^). Individuals with a BMI of 25–29.9 are considered overweight, while those with a BMI ≥ 30 are considered obese [3,4]. The global increase in obesity has had a significant impact on health and quality of life, contributing to conditions such as cardiovascular diseases, asthma, heart disease, cancer, type 2 diabetes mellitus, and dementia [5,6].

One way to overcome this problem is to take the anti-obesity or slimming drugs available in the market. Several synthetic drugs that work as slimming agents have different mechanisms, such as eliminating appetite, inhibiting fat absorption, and increasing energy expenditure [7]. Orlistat, which is shown in Figure 1, is a commonly prescribed medication for obesity management. It functions by suppressing the pancreatic lipase enzyme in the digestive tract, thereby preventing fat absorption from triglyceride decomposition [8,9]. Lipase enzymes catalyze the hydrolysis of triglycerides into free fatty acids, which are absorbed by the body and contribute to obesity, as detailed in Figure 2. Among the various types, pancreatic lipase is responsible for the hydrolysis of 50–70% of the total lipids in the body. The amino acids Ser152, Asp176, and His263 represent its catalytic site, with Ser152 playing a crucial role in lipid breakdown activity. By inhibiting pancreatic lipase, triglyceride hydrolysis is reduced, potentially decreasing the prevalence of obesity [10,11,12,13,14]. However, long-term consumption of orlistat can lead to side effects such as nausea, diarrhea, and frequent bowel movements [15,16].

The investigation has revealed that the reduction in intra-abdominal fat content correlates with a reduction in lipid digestion; when the pancreatic lipase is inhibited, the human body’s total cholesterol concentration decreases. Thus, pancreatic lipase inhibitors are candidates for medications intended to help with weight loss and help to avoid obesity. Many studies have shown that plant-based substances inhibit pancreatic lipase. Plant-derived natural chemicals are identified by their structural variety, low toxicity, and an abundance of sources. In terms of health, natural chemical-derived inhibitors are more relevant than chemically manufactured drugs, and consequently, the identification of pancreatic lipase inhibitors from plants with less side effects is an exciting field of study. Polyphenols, flavonoids, saponins, tannins, terpenoids, alkaloids, and other active compounds are found in major natural products of lipase inhibitors [15,17,18,19,20,21].

Indonesia is known as a biodiverse country with a diversity of biological resources, including medicinal plants. Indonesian people traditionally use medicinal plants for slimming, weight loss, appetite control, and as stomach and digestive medicine [22,23,24]. These plants are reported to have inhibitory activity against the action of the lipase enzyme in vitro. Secondary metabolites in these plants are known to act as inhibitors of the lipase enzyme.

**Figure 2 cimb-47-00039-f002:**
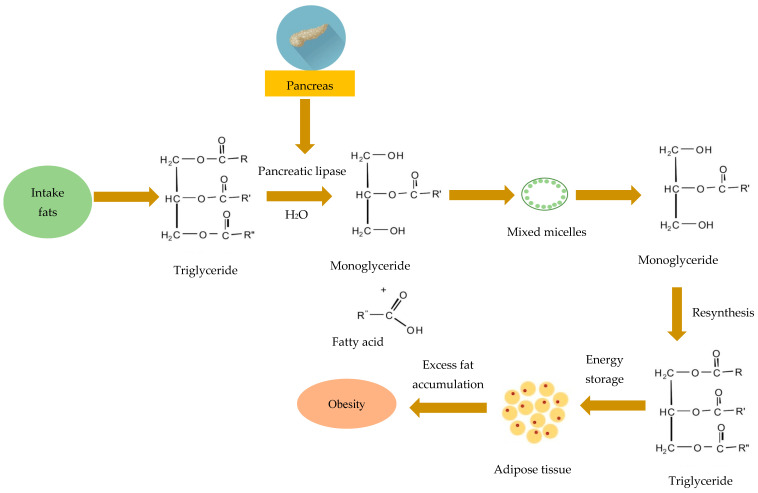
Catalysis reaction of pancreatic lipase enzyme (Modification [25]. License number: 5924750703740).

In this study, we conducted a literature review of Indonesian medicinal plants which are empirically used for slimming by Indonesian people and have been reported to inhibit the activity of the lipase enzyme in vitro, including kunci pepet (*Kaempferiae angustifolia* Rosc.), asam gelugur (*Garcinia atroviridis*), temulawak (*Curcuma xanthorrhiza*), jombang (*Taraxacum officinale* F. H. Wigg), pegagan (*Centella asiatica*), and pala (*Myristica fragrans*). This study also summarizes the content of the compounds or groups of compounds contained by these plants which are thought to have the potential, or have been proven, to be lipase inhibitors. The compounds we present in this review are based on compounds that have been reported in the referenced literature. In addition, information regarding slimming mechanisms both in vitro and in vivo is also presented.

## 2. Materials and Methods

A literature search was conducted, with keywords such as “anti-obesity”, “pancreatic lipase enzyme”, “Indonesian medicinal plants”, “in vitro”, and “in vivo” in major databases, including Scopus, Google Scholar, and PubMed.

## 3. Kunci Pepet (*Kaempferia angustifolia* Rosc.)

*Kaempferia angustifolia* is a plant that belongs to the family Zingiberaceae. *K. angustifolia* that grows abundantly in the Southeast Asian region, mainly in Indonesia [26,27,28,29], where it is commonly known as kunci pepet. The pharmacological activity of *K. angustifolia* has been widely reported as an antioxidant, weight loss agent, cough medicine, anticancer agent, and antimicrobial agent [26,30,31].

Phytochemical analysis of *K. angustifolia* extract has consistently identified the presence of flavonoids, alkaloid saponins, tannins, and terpenoids, as presented in Table 1. Pradono et al. [32] used water extract of *K. angustifolia* at the concentration of 100–300 ppm (50 ppm interval) to inhibit the action of the pancreatic lipase enzyme (Table 2). The water extract had the highest inhibitory power of 65.1% at a concentration of 200 ppm. The reported total flavonoid content was reported as 0.02 × 10^−2^% (*w*/*w*). Meanwhile, Rafi et al. [30] stated that 50% ethanol extract of *K. angustifolia* (100 µg/mL) inhibited the action of the pancreatic lipase enzyme by 59.81 ± 1.00%, as detailed in Table 3. These data strengthen the finding that *K. angustifolia* strongly inhibits the activity of the pancreatic lipase enzyme. This is thought to be due to the presence of biomarker compounds in *K. angustifolia* extracts.

The characteristics of secondary metabolites contained in the 50% ethanol extract from *K. angustifolia* are the presence of phenolics such as *o*-Coumaric acid, benzoic acid, and steroids, such as stigmasterol (Figure 3). The compound has been reported as a lipase enzyme inhibitor and as an anti-obesity agent [30,35,36,37].

The aqueous extract of hulled barley contains coumaric acid, which efficiently suppresses adipocyte development. Within a concentration of 50 mg/kg, this extract inhibited body weight gain and fat mass increase in obese male mice fed a high-fat diet [38]. This shows that coumaric acid functions strongly as an anti-obesity compound.

Flavokawain A, which is shown in Figure 4, has been isolated from *K. angustifolia* by Hanif et al. [39]. At 10 µg/mL, the compound significantly suppressed triglyceride accumulation in 3T3-L1 murine pre-adipocytes, signifying the potential as an anti-obesity agent. The synthetic form of flavokawain A effectively reduced triglyceride accumulation and blocked the development of murine pre-adipocytes (EC_50_ = 64.4 µM and IC_50_ = 74.8 µM).

Hidaka and colleagues [40] utilized *K. parviflora*, a member of the same genus Kaempferia, in an in vivo study to treat obesity. Tsumura Suzuki obese diabetes (TSOD) mice were utilized as an overweight model. The ethanolic extract of *K. parviflora* lowered mouse body weight and the thickness of the subcutaneous fat layer more than the PMF fraction, which is commonly used as a nutritional supplement to manage obesity-related skin problems.

## 4. Asam Gelugur (*Garcinia atroviridis*)

Asam gelugur is a fruit from the Guttiferae family, specifically the *Garcinia* genus distributed in tropical Asia. This fruit is cultivated extensively in Aceh and Sumatera Island, Indonesia. In North Sumatera, *Garcinia atroviridis* is often utilized as a food component [41,42,43,44,45].

Several studies have reported that *G. atroviridis* extract contains flavonoids, saponins, tannins, steroids, and terpenoids, as presented in Table 4. Iswantini et al. [46] adopted methanol as a solvent to extract *G. atroviridis,* followed by fractionation using *n*-hexane, butanol, and dichloromethane. The butanol fraction produced the largest yield of 8.24% and the greatest inhibitory power, with an IC_50_ of 46.91 µg/mL. This fraction was subjected to further partitioning through column chromatography, utilizing an eluent comprising a mixture of chloroform, with methanol ranging between 65:35 and 55:45. Fraction E of the butanol fraction provided the highest inhibitory power of 52.89% at a concentration of 10 µg/mL. The fraction is believed to contain hydroxy citric acid (HCA) (Figure 5), a compound abundant in *G. atroviridis* and which can inhibit pancreatic lipase enzyme activity. In the Krebs cycle, HCA inhibits the formation of ATP (adenosine 5′-triphosphate) citrate lyase (ACL), an enzyme that converts citric acid into acetyl coenzyme A (CoA). The coenzyme is converted to malonyl co-A, as well as fatty acids and cholesterol. Therefore, the HCA contained in *G. atroviridis* plays an active role in inhibiting fat formation [47,48].

In an in vivo study, the composition of water extract of *G. atroviridis* fruit and 50% ethanol extract of *K. angustifolia,* in a ratio of 2:1, effectively reduced the body weight of fattened mice by approximately 12.6%, at a dose of 13.44 mg/20 g BW [52]. In another study, Achmadi [53] applied *G. atroviridis* in an in vivo investigation on rats. Over two weeks of daily treatment with 2 mL of a 2% potassium solution, the study recorded a decreased body weight, enhanced HDL (high-density lipoprotein) concentrations from 35 to 63 mg/day, and decreased blood cholesterol LDL (low-density lipoprotein) levels from 63 mg/day to 50 mg/day. The results showed the ability of *G. atroviridis* extract to lower body weight and prevent the production of cholesterol. In addition, Lim et al. [51] used a methanol extract of *G. atroviridis* to reduce the body weight of female and male rats with an oral lethal dose (LD_50_) of 2000 mg/kg for nine weeks of treatment.

The anti-obesity effect of HCA can also be addressed through serotonin regulation [45]. In human therapeutic trials, Preuss et al. [54] stated that HCA reduced hunger, weight loss, and plasma leptin levels, while increasing serum serotonin levels and improving lipid profile. In addition, Kim et al. [55] indicated that the potential of HCA to reduce body weight increase was possibly related to its combined effects on the metabolic and serotonin pathways.

## 5. Temulawak (*Curcuma xanthorrhiza*)

Temulawak, or Java turmeric (*Curcuma xanthorrhiza* Roxb), belongs to the Zingiberaceae family, and has been traditionally utilized as a component in jamu, an Indonesian herbal drink and therapy. This medicinal plant is predominantly cultivated in Indonesia and other Southeast Asian nations. Some species of the genus Curcuma were also discovered in Australia and the southern Pacific area. *C. xanthorrhiza* contains many phenolic compounds, including curcuminoid, as well as terpenoids, such as xanthorrhizol, *α*-curcumin (sesquiterpene), *β*-curcumin (sesquiterpene), and camphor (monoterpene). It also contains amino acids, such as lysine, alanine, valine, and tryptophan [56,57,58,59,60,61].

Several investigations showed that the phytochemical analysis of *C. xanthorrhiza* extract was positive for flavonoids, alkaloids, saponins, tannins, steroids, and terpenoids (Table 5). Curcuminoid in *C. xanthorrhiza* belongs to the diarylheptanoids group (Figure 6a) and consists of curcumin, demethoxycurcumin, and bisdemethoxycurcumin [62]. The difference between these compounds was observed from the alkyl groups R1 and R2. Curcumin is characterized by R1 = OMe and R2 = OMe, while demethoxycurcumin had R1= OMe and R2= H. The compound bisedemethoxycurcumin featured R1 = H and R2 = H (Figure 6b–d) [63].

The ability of curcumin to prevent obesity can be related to its inhibition of adipocyte differentiation. Adipocytes are formed when mesenchymal cells differentiate into preadipocytes, which eventually mature into adipocytes. Over-normative adipogenesis results in an excessive buildup of adipocytes, which contributes to obesity. Curcumin has the potential to influence the life cycle of adipocytes by suppressing preadipocyte proliferation and mitogenesis, inhibiting adipogenesis, and inducing mature adipocyte apoptosis. Kim et al. [67] reported that curcumin inhibits adipogenesis in both 3T3-L1 mouse cells and human primary preadipocytes, as demonstrated by a study of the intracellular storage of lipids. Additionally, through an in vivo study, in animal models of diet-induced obesity, Ejaz et al. [68] revealed that curcumin can also prevent obesity by inhibiting adipocyte differentiation.

Other research has been conducted by Batubara et al. [66]. They stated that ethyl acetate of *C. xanthorrhiza* inhibited 80.5% of the activity of pancreatic lipase enzyme at a concentration of 500 µg/mL. Meanwhile, at the same concentration, methanol and *n*-hexane extract provided inhibitions of 5.90% and 36.09%, respectively. In addition, Oon et al. [69] revealed that xanthorrhizol, isolated from *C. xanthorrhiza* extract, reduced 27% cholesterol levels in HT29 colon cells, and adipogenesis in 3T3-L1 at a concentration of 15 µg/mL. According to Kim et al. [70], 50 mg/kg *C. xanthorrhiza* extract and 15 mg/kg xanthorrhizol reduced body weight and epidydimal fat mass on cancer-induced adipose tissue wasting in CT26 tumor-bearing mice. These findings resulted that xanthorrhizol (Figure 7) contained in *C. xanthorrhiza* also has potential as an anti-obesity agent.

## 6. Jombang (*Taraxacum officinale* F. H. Wigg)

Dandelion (*Taraxacum officinale* F. H. Wigg), known as “Jombang” in Indonesia, is a medicinal plant belonging to the Asteraceae family [71]. The term ’Taraxacum’ originated from the Greek words ’taraxos’ and ’akos’, which represented ’disorder’ and ’remedy’, respectively. This plant is widely distributed globally and primarily grows in temperate, as well as subtropical, parts of the Northern Hemisphere. It is particularly prevalent in regions of Asia and North America [72,73,74,75].

Several investigations have shown that *T. officinale* extract contained alkaloids, flavonoids, tannins, steroids, and terpenoids, as presented in Table 6. Furthermore, triterpenoids, including taraxerol, sesquiterpenoids, and phenolic acids such as chicoric acid, have been identified. Some sterols, such as β-sitosterol and campesterol in *T. officinale*, have anti-obesity properties (Figure 8) [74,76,77,78,79].

Adia et al. [71] reported that the 96% ethanol of *T. officinale* leaves extracts at 500 ppm inhibited 103.85% of pancreatic lipase enzyme activity. Zhang et al. [81] conducted an in vitro investigation using a 95% ethanol extract of *T. officanale* at concentrations of 4, 12.5, 25, 100, 125, and 250 µg/mL. The results showed that increasing the extract concentration corresponded to a higher inhibitory effect on pancreatic lipase activity, as detailed in Table 7. An inhibitory power of 86.3% is obtained at a concentration of 250 µg/mL. At a concentration of 250 µg/mL, orlistat, a positive control, inhibited 95.7% of the enzyme’s activity. The study was further extended to in vivo experiments where 400 mg/kg of 95% ethanol extract of *T. officinale* was applied to mice. After 90 and 180 min, the incremental plasma triglyceride levels were substantially lowered. Compared to the control group (11,068 ± 2054 mg·min/dL), the *T. officinale* group (8504 ± 1950 mg·min/dL) had significantly lower areas under the response curves (AUC) for the triglyceride response. The study by Rao et al. [82] also supports the function of *T. officinale* extract as an anti-obesity drug in vivo. Male Sprague Dawley rats fed a high-fat Diet (HFD) for 10 weeks were treated with the extract at doses of 150 mg/kg and 300 mg/kg. The results showed that the intestinal absorption of fat was inhibited, leading to a reduction in body weight.

## 7. Pegagan (*Centella asiatica*)

Pegagan (*Centella Asiatica*) is a plant that belongs to the Apiaceae family. It often grows in swamp areas as well as in tropical and subtropical countries, such as Southeast Asia, North Australia, South Africa, and Eastern South America [83,84,85,86,87,88].

Table 8 shows the general content of alkaloids, flavonoids, tannins, saponins, steroids, and terpenoids in *C. asiatica*. Yunarto et al. [89] observed that ethanol extract of *C. asiatica* effectively inhibited pancreatic lipase enzyme activity, with an IC_50_ of 21.14 µg/mL. According to Hong et al. [90], the extract inhibited 68.1% of pancreatic lipase enzyme activity to prevent triglyceride hydrolysis.

In vivo studies support the effectiveness of *C. asiatica* extract in preventing obesity. Its combination with *Sauropus androgynus* extracts at a dose of 25:50 mg/kg BW inhibited lipid peroxidation in the Swiss Webster strain of mice which was induced by obesity [91]. Chang et al. [84] also reported that 300 mg/kg of *C. asiatica* extract reduced body fat deposition by modulating the development of cholesterol levels and lipid metabolism-related genes in the liver in mice with high-fat-induced obesity.

**Table 8 cimb-47-00039-t008:** Results of phytochemical analysis of *C. asiatica* extract from some studies.

Extract	Secondary Metabolite						Reference
	Alkaloids	Flavonoids	Saponins	Tannins	Steroids	Terpenoids	
Ethanol	+	+	+	+	-	Not tested	[92]
Ethanol	+	+	Not tested	+	Not tested	Not tested	[93]
Methanol	+	+	+	+	+	+	[94]
Methanol	+	+	Not tested	+	Not tested	Not tested	[93]
Acetone	+	-	-	+	-	+	[94]
Chloroform	+	+	-	-	-	-	[94]
Water	+	-	-	-	-	+	[94]
Water	+	+	Not tested	+	Not tested	Not tested	[93]
Leaves	-	+	+	+	+	+	[95]

Asiaticoside, madecassoside, asiatic acid, and madecassic acid (Figure 9) are the four largest quantities of terpenoids found in *C. asiatica*. These four compounds have been identified as hepatoprotectors because of their ability to inhibit lipopolysaccharide/D-galactosamine-induced acute liver damage [83,96,97]. Asiaticoside and madecassoside are classified as saponins (pentacyclic triterpene glycosides), whereas the corresponding aglycones (sapogenins) are asiatic acid and madecassic acid. The biological activity of saponins is attributed to a hydrophilic sugar chain, called glycone. It is produced by the isoprenoid pathway, and is a component of the hydrophobic triterpenoid structure aglycone. The most prevalent pentacyclic triterpenoids in *C. asiatica* are saponins and their aglycones [98].

Sun et al. [99] reported that madecassoside inhibits weight gain and lipogenesis through stimulation of the expression-activated protein kinase (AMPK) in mesenteric fat and epididymal fat. Additionally, Uddandrao et al. [100] determined that asiatic acid produced from *C. asiatica,* at a concentration of 20 mg/kg BW, might suppress adipogenesis by lowering inflammation in adipose tissue when applied to high-fat diet-induced obesity in the Sprague Dawley rat model for 42 days.

## 8. Pala (*Myristica fragrans*)

Myristica *fragrans,* the scientific name for nutmeg, is commonly referred to as “pala” in Indonesian. This plant belongs to the family of Myristicaceae. It originates from Maluku, Indonesia, or the Spice Islands. *M. fragrans* can be found in Southeast Asia and other nations, including Indonesia, Malaysia, Vietnam, India, and Sri Lanka. The tree is a fragrant evergreen with scattered branches that stands 9–12 m tall. The dried seed of *M. fragrans* has a sweet flavor and a unique, pleasant smell [101,102].

Phytochemical analysis of *M. fragrans* shown that the extract of *M. fragrans* contained alkaloids, flavonoids, saponins, terpenoids, and tannins (Table 9). The essential oils contained in nutmeg are myristicin, limonene, terpine-4-ol, β-pinene, and α-pinene. Ethnomedically, *M. fragrans* has been recognized to help with digestion, stomach discomfort, diarrhea, spleen and sore throats, allergy relief, and analgesic complications. Pharmacologically, *M. fragrans* exhibits antidiabetic, anti-inflammatory, antioxidant, and antibacterial properties [102,103,104].

Yakaiah et al. [107] reported that an ethanol extract of *M. fragrans* seeds could inhibit 66.24% of pancreatic lipase action, while orlistat as a positive control inhibited 81.57%. Tetrahydrofuran the most abundant compound in the ethanol extract of *M. fragrans* seed, has the binding energy of −6.2 kcal/mol, while orlistat has the binding energy of −7.56 kcal/mol. Nguyen et al. [108] also investigated whether the tetrahydrofuran (Figure 10) contained in *M. fragrans* extract can reduce adipose tissue mass and body weight in a diet-induced animal model.

## 9. Conclusions and Prospects

Obesity can be treated by inhibiting the pancreatic lipase enzyme. This enzyme breaks down dietary lipids into monoglycerides and fatty acids, making them easier to absorb and distribute throughout the body. Kunci pepet (*Kaempferiae angustifolia* Rosc.), asam gelugur (*Garcinia atroviridis*), temulawak (*Curcuma xanthorrhiza*), jombang (*Taraxacum officinale* F. H. Wigg), pegagan (*Centella asiatica*), and pala (*Myristica fragrans*) are the suggested traditional Indonesian medicinal plants. The biomarker compound reported in vitro to inhibit the activity of the lipase enzyme is coumaric acid contained in kunci pepet (*K. angustifolia*); hydroxycitric acid in asam gelugur (*G. atroviridis*); curcumin and xanthorrhizol in temulawak (*C. xanthorrhiza*); taraxerol in jombang (*T. officinale*); asiatic acid, madecassic acid, asiaticoside, and madecassoside in pegagan (*C. asiatica*); and tetrahydrofuran in pala (*M. fragrans*). Furthermore, we also found in vivo evidence that these biomarker chemicals can reduce triglyceride formation in adipose tissue in obese male animal models, such as mice and rats. These findings highlighted the importance of these six medicinal herbs with useful anti-obesity potential.

The development of a synergistic combination of these six plants in the future for in vitro and in vivo treatment as an anti-obesity agent can be carried out further. Our research team has successfully combined the extract of kunci pepet (*K. angustifolia*) and asam gelugur (*G. atroviridis*) as an anti-obesity combination, based on the in vitro assay of pancreatic lipase enzyme inhibition and the in vivo assay using a mouse as a test animal model. This combination has obtained a granted patent (IDP000051408) from the Ministry of Law and Human Rights of The Republic of Indonesia. Based on the results of our research, Flavokawain A has been found to be the active compound of *K. angustifolia*. Now, our research team will continue to carry out clinical assay to prove weight loss in respondents. Therefore, it is necessary to keep investigating combinations of medicinal plant extracts, and there are promising future prospects for investigating more medicinal plants as anti-obesity medicines.

However, the challenge that may be faced in developing herbal medicines is the involvement of all stakeholders in developing herbal medicines that are safe to consume for obesity sufferers and maintain their efficacy as anti-obesity products.

## Figures and Tables

**Figure 1 cimb-47-00039-f001:**
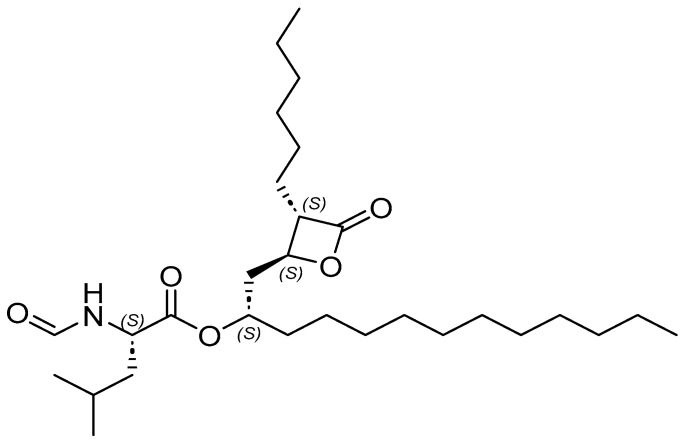
The chemical structure of orlistat (PubChem ID: 3034010).

**Figure 3 cimb-47-00039-f003:**
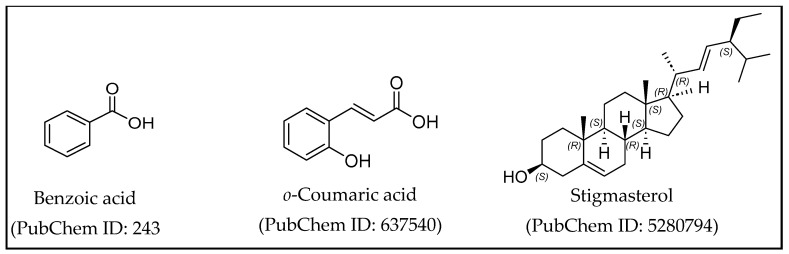
The chemical structures of benzoic acid, *o*-Coumaric acid, and stigmasterol.

**Figure 4 cimb-47-00039-f004:**
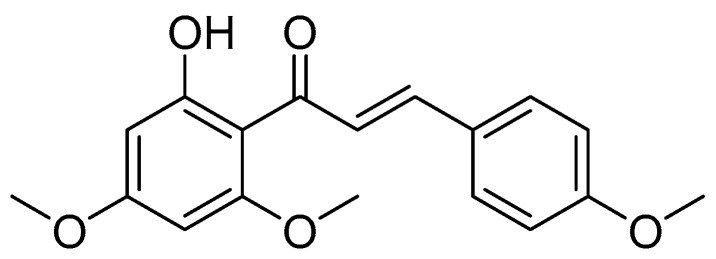
The chemical structure of flavokawain A (PubChem ID: 5355469).

**Figure 5 cimb-47-00039-f005:**
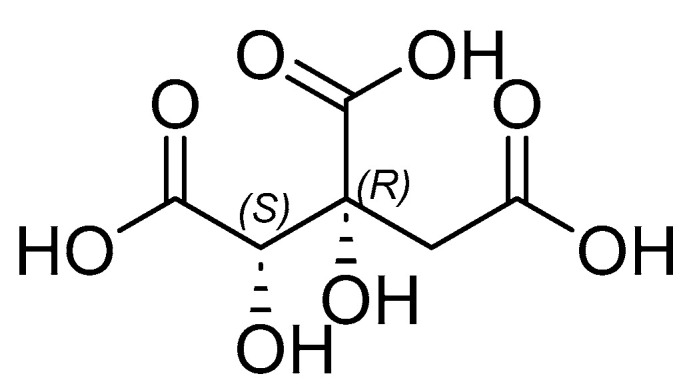
The chemical structure of hydroxycitric acid (HCA). (PubChem ID: 123908).

**Figure 6 cimb-47-00039-f006:**
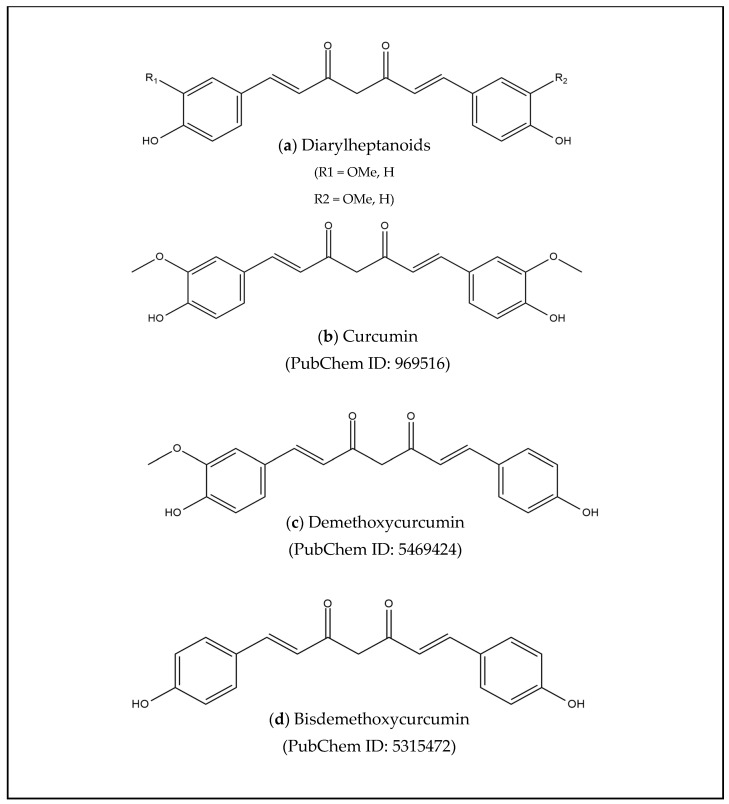
The chemical structures of diarylheptanoids, curcumin, demethoxycurcumin, and bisdemethoxycurcumin.

**Figure 7 cimb-47-00039-f007:**
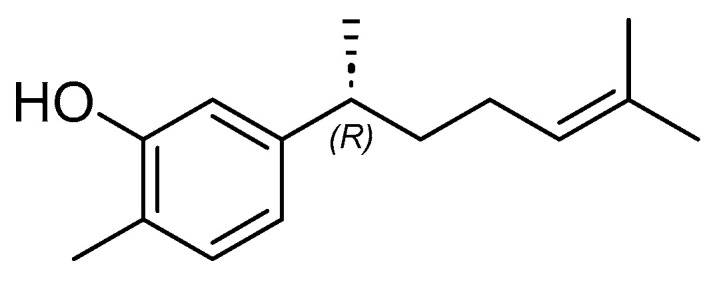
The chemical structure of xanthorrhizol (PubChem ID: 93135).

**Figure 8 cimb-47-00039-f008:**
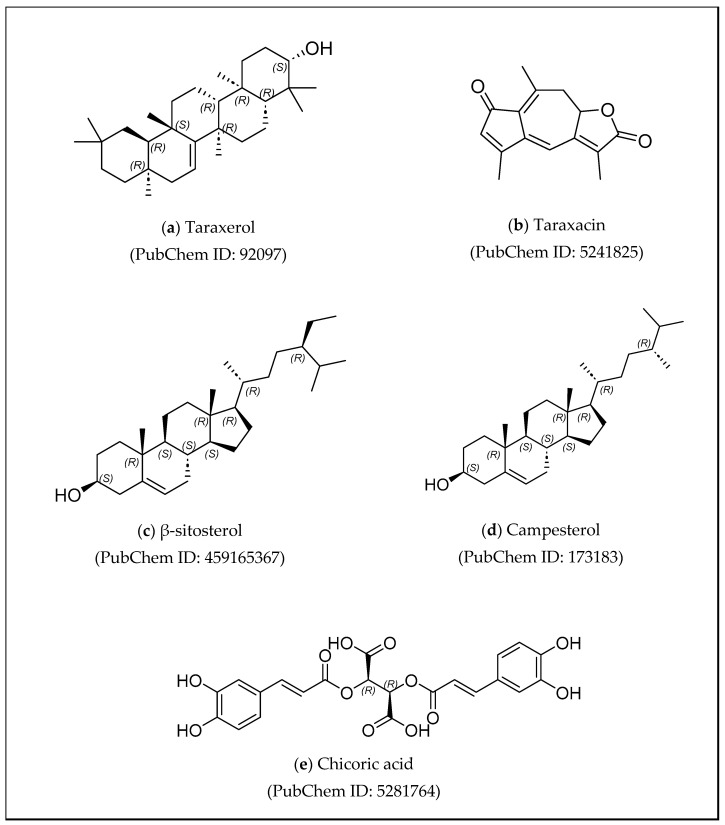
Structure of taraxerol, taraxacin, β-sitosterol, campesterol, and chicoric acid.

**Figure 9 cimb-47-00039-f009:**
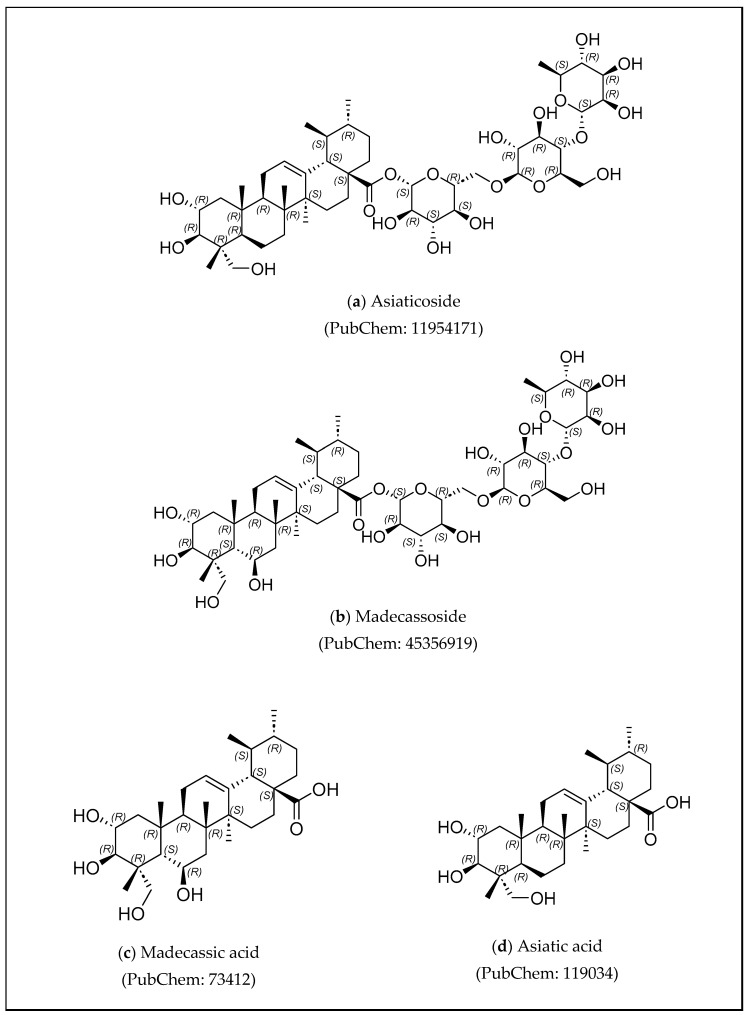
Structure of asiaticoside, madecassoside, madecassic acid, and asiatic acid.

**Figure 10 cimb-47-00039-f010:**
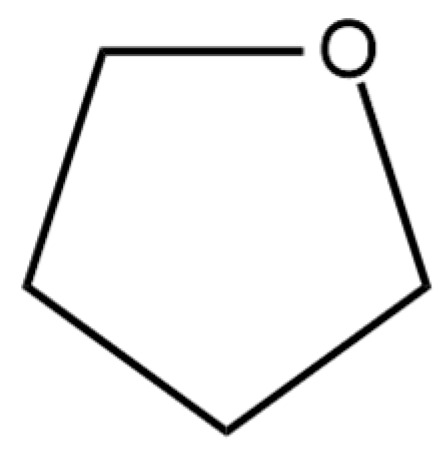
Structure of tetrahydrofuran.

**Table 1 cimb-47-00039-t001:** Results of phytochemical analysis of *K. angustifolia* extract from some studies.

Extract	Secondary Metabolite						Reference
	Alkaloids	Flavonoids	Saponins	Tannin	Steroids	Terpenoids	
Water	+	+	+	-	-	-	[33]
70% ethanol	+	+	+	-	-	+	[33]
Methanol	-	-	+	-	Not tested	+	[34]

**Table 2 cimb-47-00039-t002:** Percentage of pancreatic lipase enzyme inhibition from *K. angustifolia* water extract [32].

Concentration (ppm)	Inhibition Percentage (%)
100	10.8
150	56.2
200	65.1
250	36.9
300	26.6

**Table 3 cimb-47-00039-t003:** Percentage of pancreatic lipase enzyme inhibition from *K. angustifolia* extract (100 µg/mL) with different solvents [30].

Extract	Inhibition Percentage (%)
Orlistat	72.54 ± 2.04
30% ethanol	25.92 ± 0.84
50% ethanol	59.82 ± 1.00
70% ethanol	31.32 ± 2.61
96% ethanol	30.66 ± 0.90
Water	34.55 ± 0.42

**Table 4 cimb-47-00039-t004:** Results of phytochemical analysis of *G. atroviridis* extract from some studies.

Extract	Secondary Metabolite						Reference
	Alkaloids	Flavonoids	Saponins	Tannins	Steroids	Terpenoids	
70% ethanol	-	+	+	-	-	-	[49]
70% ethanol	-	+	-	+	+	+	[50]
Methanol	-	+	+	-	-	+	[51]

**Table 5 cimb-47-00039-t005:** Results of phytochemical analysis of *C. xanthorrhiza* extract from some studies.

Extract	Secondary Metabolite						Reference
	Alkaloids	Flavonoids	Saponins	Tannins	Steroids	Terpenoids	
Ethanol	+	+	-	-	+	-	[64]
70% ethanol	-	+	+	+	Not tested	Not tested	[65]
96% ethanol	+	+	-	-	-	+	[66]

**Table 6 cimb-47-00039-t006:** Results of phytochemical analysis of *T. officinale* extract from some studies.

Extract	Secondary Metabolite						Reference
	Alkaloids	Flavonoids	Saponins	Tannins	Steroids	Terpenoids	
96% ethanol	+	+	-	+	+	+	[71]
80% ethanol	+	+	-	-	Not tested	-	[80]

**Table 7 cimb-47-00039-t007:** Inhibitory power of 95% ethanol extract of *T. officinale* on the activity of the pancreatic lipase enzyme [81].

Extract	Concentration (µg/mL)	% Inhibition	IC_50_ (µg/mL)
95% ethanol of *T. officinale*	4	8.1	78.2
	12.5	21.9	
	25	34.1	
	100	56.5	
	125	63.0	
	250	86.3	
Orlistat	4	90.4	0.22
	250	95.7	

**Table 9 cimb-47-00039-t009:** Results of phytochemical analysis of *M. fragrans* seed extract from some studies.

Extract	Secondary Metabolite						Reference
	Alkaloids	Flavonoids	Saponins	Tannins	Steroids	Terpenoids	
Ethanol	+	+	+	-	Not tested	+	[105]
Ethanol	+	+	+	+	Not tested	+	[106]
Ethyl acetate	-	+	+	-	Not tested	-	[106]

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
