# Peer review of "The Strong Inhibition of Pancreatic Lipase by Selected Indonesian Medicinal Plants as Anti-Obesity Agents"

_cimb, 2025, doi:10.3390/cimb47010039_

Round 1
Reviewer 1 Report
Comments and Suggestions for Authors
In the manuscript titled the strong inhibition of pancreatic lipase by selected indonesian medicinal plants as antiobesity agents, this study presents an interesting review on the potential of selected Indonesian medicinal plants to inhibit pancreatic lipase, with implications for anti-obesity treatment. After carefully reading the manuscript, I don't think the article meets the basic requirements for a review article writing. I provide the following comments:
1. In five sections investigating the anti-obesity effects of medicinal plants in Indonesia, there is a lack of a comprehensive evaluation of various viewpoints.
2. The conclusion section lacks an in-depth summary of the main body of the article
3. The conclusion section lacks the presentation of issues in the current research.
Overall, the article needs to be reorganized to reflect logic and systematicity.
Author Response
Comments 1: In five sections investigating the anti-obesity effects of medicinal plants in Indonesia, there is a lack of a comprehensive evaluation of various viewpoints. |
Response 1: Thank you for pointing this out. We agree with this comment. Therefore, we have revision: the introduction (page 2 lines 32-35; 45-48; 64-89); the detailed of how coumaric acid can be as antiobesity in other researches (page 2 lines 182-185); the effect of HCA as antiobesity (page 3 lines 251-256); the ability curcumin as antiobesity (page 5 lines 299-309); asiaticoside and madecassoside as antiobesity (page 10 lines 489-498); and we have one plant, pala as antiobesity (page 10 lines 502-528). We hope each revision that we have added can be more comprehensive review.
|
Comments 2: The conclusion section lacks an in-depth summary of the main body of the article. |
Response 2: Thank you for the suggestions. We agree with this suggestion. We have added more conclusions as related to the main body of review (page 11-18 lines 530-550).
|
Comments 3: The conclusion section lacks the presentation of issues in the current research. Response 3: Thank you for the suggestions. We agree with this suggestion. We have added more conclusions and future prospects related to the current research and point of view of Indonesian medicinal plants as antiobesity (page 11-18 lines 530-550).
4. Response to Comments on the Quality of English Language |
Point 1: |
Response 1: Thank you for the comments. We agree of this comment. We have applied proofreading by GoodLingua of our manuscript to increase and revision the grammar in the manuscript. |

Reviewer 2 Report
Comments and Suggestions for Authors
Dear Editors,
several plants in the last years have demonstrated anti obesity properties and their contents, are studied and are under consideration in clinical trials to be approved. In this regard, your study could lack, and lacks of originality. An additional new article review on the topic, but specifically focused on selected Indonesian Medicinal Plants could be welcome, but it should be larger and more complete tahn the present one, which cannot claimed as a Review, at most a mini review. It should be well written, without scientific errors, well organized, with a sufficinet number of references.
On the contrary:
References are poor. 72 references are limited for a review. Ate lease 100-150 references are needed.
The Eglish language is very poor. There are a lot of grammar erros along all manuscript (plural in place of singular, wrong verbs, prepositions missing, sentences not correctly organized). The manuscript needs the revision of an English expert.
There are scientific incorrectenesses such as "Body Mass Index (BMI), which is computed by 35 dividing body weight in kilograms (kg) by body height (m2 )."m2" is wrong "cm" is correct.
References in the text have to be reported only as numbers in square breakets. Please remove citations by names and date.
Please, report all manuscrip on a 2024 template.
Conclusions need improvement.
Authors should provide permission by authors/journal of Figures used.
Comments on the Quality of English Language
Need extensive improvement
Author Response
Comments 1: References are poor. 72 references are limited for a review. Ate lease 100-150 references are needed. |
Response 1: Thank you for the suggestion. We agree with this suggestion. We have added more references which obtained up to 110 references related to our review (page 11-18; lines 561-837). We hope it can increase the comprehensive of our review.
|
Comments 2: There are scientific incorrectenesses such as "Body Mass Index (BMI), which is computed by 35 dividing body weight in kilograms (kg) by body height (m2 )."m2" is wrong "cm" is correct. |
Response 2: Thank you for the comment. According to World Health Organization (https://www.who.int/news-room/fact-sheets/detail/obesity-and-overweight) and National Library of Medicine (https://www.ncbi.nlm.nih.gov/books/NBK279167/), obesity specifications are based on Body Mass Index (BMI), which is computed by dividing body weight in kilograms (kg) by body height (m2). Therefore, we still use m2 for calculation BMI. |
Comments 3: References in the text have to be reported only as numbers in square breakets. Please remove citations by names and date. Response 3: Thank you for the correction. We have removed citations by names and date. We can see in highlight yellow, such as in Table 1 lines 154-155; Table 4 lines 225-226; Table 5 lines 284-286; Table 6 lines 381-382.
Comments 4: Please, report all manuscript on a 2024 template. Response 4: Thank you for the suggestion. However, we used the template from invitation manuscript of Ms. Norah Tang.
Comments 5: Conclusions need improvement. Response 5: Thank you for the suggestions. We agree with this suggestion. We have added more conclusions and future prospects related to the current research and point of view of Indonesian medicinal plants as anti-obesity (page 11-18 lines 530-550).
Comments 6: Authors should provide permission by authors/journal of Figures used. Response 6: Thank you for the correction. We have provided permission by journal and we have license number such as in Figure 1 (page line 153).
4. Response to Comments on the Quality of English Language |
Point 1: The English language is very poor. There are a lot of grammar erros along all manuscript (plural in place of singular, wrong verbs, prepositions missing, sentences not correctly organized). The manuscript needs the revision of an English expert. |
Response 1: Thank you for the suggestion. We have provided proofreading by GoodLingua to increase the grammar in English. We hope it can be well received. |

Reviewer 3 Report
Comments and Suggestions for Authors
The reviewer's comments file has been uploaded

Author Response
Comments 1: In the introduction, it would be beneficial to discuss the drugs currently used as anti-obesity agents and their associated side effects. This would help emphasize the review's purpose: to present natural extracts from Indonesian plants as potential anti-obesity agents with minimal side effects. |
Response 1: Thank you for the suggestions. We agree with this suggestion. We have added the associated minimal side effects when using Indonesian medicinal plants as anti-obesity. We can observe it in page 2 lines 32-35; 45-48; 64-89.
|
Comments 2: In paragraphs 3 to 7, the results of the phytochemical analyses and biochemical studies of the extracts from various medicinal plants are presented in a list-like format. These sections need to be enhanced to make the reading more engaging and fluid. Additionally, the atomic numbering of Figures 5 and 7 requires correction. |
Response 2: Thank you for the comments. We agree with this comment. However, we have made the revisions with these sections and we have added how biomarker from plants can be as antiobesity, such as how coumaric acid can be as anti-obesity in other (page 2 lines 182-185); the effect of HCA as antiobesity (page 3 lines 251-256); the ability curcumin as antiobesity (page 5 lines 299-309); asiaticoside and madecassoside as antiobesity (page 10 lines 489-498); and we have one plant, pala as antiobesity (page 10 lines 504-530). We hope each revision that we have added can be more comprehensive review. Thank you for the correction to Figures 5 and 7. We have analyzed the atomic numbers via PubChem and we have adjusted the atomic numbering of these figures.
|
Comments 3: Considering these factors, I recommend revising the text before it is published. Response 3: Thank you for the comments. We agree with this comment. We have tried to revised and enhanced the main body our manuscript, we have added more references which related to the review, and we have increased the grammar via proofreading by GoodLingua. We hope it can be more comprehensive to revise the manuscript.
4. Response to Comments on the Quality of English Language |
Point 1: |
Response 1: Thank you for the suggestion. We have provided proofreading by GoodLingua to increase the grammar in English. We hope it can be well received. |
|
|

Round 2
Reviewer 2 Report
Comments and Suggestions for Authors
Dear Authors,
Thanks for your revisions which have enhanced the quality of your manuscript.
Anyway, two points remain not addressed. Concerning point 2, I didn't explain myself well. The formula is correct, but your explanation of the formula is wrong. You say, "weight in kilograms divided by the height of the individual". It is not correct! Instead, it is a matter of dividing by the height in meters raised to the power of 2. Please correct.
Pont 4. It does not matter which template was provided to you by who invited you, the current template is available in the instructions for authors and therefore that should be used. Please, pour your work into the correct template.
A new improvement, which I noticed, rereading your article is however necessary. The authors discuss different bioactive molecules of vegetal origin, most of which are characterized by the presence of multiple chiral carbons whose precise stereochemistry is fundamental for their activity. Often, the same molecule, but with different stereochemistry (enantiomer or diastereoisomer) is inactive or has different activity. Therefore, it is of great importance to highlight it on molecules with stereogenic centers. Therefore, the authors should insert the correct stereochemical descriptors on all chiral carbons of optically active molecules present in their work. Please, do it in Figures 9, 8, 7, 5, 3, and 1. In almost all Figures, only the descriptors are missing, in others it is necessary indicate also the stereochemistry (Figure 5).
In Figure 6a, it is necessary specifying R1 and R2.
The structure of Taraxacin is wrong. Please, revise.
In conclusion, even if some criticisms have been addressed during the first round revision, several other ones came out on a second reading, and my fear is that probably on a third reading other ones could emerge, confirming that the work is really not very robust and not well done. I therefore advise the authors to satisfy my major revisions but before resubmitting it, to check carefully that there are no other shortcomings.
Author Response
Comments 1: Anyway, two points remain not addressed. Concerning point 2, I didn't explain myself well. The formula is correct, but your explanation of the formula is wrong. You say, "weight in kilograms divided by the height of the individual". It is not correct! Instead, it is a matter of dividing by the height in meters raised to the power of 2. Please correct. |
Response 1: Thank you for the correction. We agree with this correction. We have reapired as the correction, becomes: The WHO defines obesity based on Body Mass Index (BMI), computed by dividing body weight in kilograms (kg) by height2 (m2) (Page 1 lines 39-40).
|
Comments 2: Pont 4. It does not matter which template was provided to you by who invited you, the current template is available in the instructions for authors and therefore that should be used. Please, pour your work into the correct template |
Response 2: Thank you for the correction. We agree with this correction. We have used manuscript on a 2024 template. |
Comments 3: A new improvement, which I noticed, rereading your article is however necessary. The authors discuss different bioactive molecules of vegetal origin, most of which are characterized by the presence of multiple chiral carbons whose precise stereochemistry is fundamental for their activity. Often, the same molecule, but with different stereochemistry (enantiomer or diastereoisomer) is inactive or has different activity. Therefore, it is of great importance to highlight it on molecules with stereogenic centers. Therefore, the authors should insert the correct stereochemical descriptors on all chiral carbons of optically active molecules present in their work. Please, do it in Figures 9, 8, 7, 5, 3, and 1. In almost all Figures, only the descriptors are missing, in others it is necessary indicate also the stereochemistry (Figure 5). Response 3: Thank you for the correction. We have added the stereochemical descriptors on all chiral carbons of optically active molecules present in their work. The chirality of these compounds is not described in the reference literature that we cited. We added chirality of compounds based on PubChem data and our chirality knowledge.
Comments 4: In Figure 6a, it is necessary specifying R1 and R2. Response 4: Thank you for the correction. We have made the specifying R1 and R2 in Figure 6a (Page 4, line 289).
Comments 5: The structure of Taraxacin is wrong. Please, revise. Response 5: Thank you for the correction. We agree with this suggestion. We have repaired the structure of Taraxacin (Page 6, line 354).
Comments 6: In conclusion, even if some criticisms have been addressed during the first round revision, several other ones came out on a second reading, and my fear is that probably on a third reading other ones could emerge, confirming that the work is really not very robust and not well done. I therefore advise the authors to satisfy my major revisions but before resubmitting it, to check carefully that there are no other shortcomings. Response 6: Thank you for the suggestion. We agree with this suggestion. We have tried our best description to revise the conclusion (page 11 lines 543-554). We hope our conclusion can be well received.
|

Reviewer 3 Report
Comments and Suggestions for Authors
The reviewer's comments have been uploaded

Author Response
Comments 1: The review titled "The Strong Inhibition of Pancreatic Lipase by Selected Indonesian Medicinal Plants as Antiobesity Agents" has been revised and enhanced. It now includes the phytochemical and pharmacological characteristics of the medicinal plant Myristica fragrans, which were not covered in the initial version of the manuscript. |
Response 1: Thank you for the appreciation. We have tried to revised and enhanced our manuscript regarding the comments and suggestions of distinguished reviewers.
|
Comments 2: I recommend adding a "Materials and Methods" section to inform readers about the databases used by the authors. |
Response 2: Thank you for the suggestion. We agree with this suggestion. We have added “Materials and Methods” in the manuscript (Page 3, lines90-93).
|
Comments 3: Taking these factors into account, I recommend that the manuscript be published. Response 3: Thank you for the recommendation. We hope our manuscript can be published in the journal of CIMB.
|

Round 3
Reviewer 2 Report
Comments and Suggestions for Authors
Dear Authors,
maybe we misunderstood each other again, but I asked that the correct stereochemical descriptors be added to all chiral carbons in the chemical structures that you entered in the manuscript. I do not see any in the latest version of the work provided by the system.
Author Response
Comments 1: Dear Authors, maybe we misunderstood each other again, but I asked that the correct stereochemical descriptors be added to all chiral carbons in the chemical structures that you entered in the manuscript. I do not see any in the latest version of the work provided by the system |
Response 1: Thank you for the correction and suggestions. We agree with this correction. We have added to all chiral carbons in the chemical structures in the manuscript. Such as Figure 1 (page 1 lines 59-62); Figure 3 stigmasterol (lines 183-192); figure 5 (lines 245-253); figure 7 (lines 304-310); figure 8 taraxerol, β-sitosterol, Campesterol, Chicoric acid (lines 346-376); figure 9 (lines 444-482). We hope it can be well received. |
|
|

Round 4
Reviewer 2 Report
Comments and Suggestions for Authors
Now my request has been addressed. Accepted.